# A Modified Green-Ampt Model and Parameter Determination for Water Infiltration in Fine-textured Soil with Coarse Interlayer

**Shuai Chen [1], Xiaomin Mao [1,\*] and Chunying Wang [2]**

[1] College of Water Resources & Civil Engineering, China Agricultural University, Beijing 100083, China; slsdchen@163.com

[2] School of Water Conservancy, North China University of Water Resources and Electric Power, Zhengzhou 450045, China; wangchunying1987@yahoo.com

\* Correspondence: maoxiaomin@cau.edu.cn; Tel.: +86-010-6273-8498

**Abstract:** A modified Green-Ampt model was developed to simulate water infiltration in fine-textured soil with a coarse interlayer. Because under such a soil structure, the two soils may not be fully saturated during infiltration, the model introduced two parameters—that is, the saturation coefficients *a* and *b*, to reflect the incomplete saturation condition and their influence on infiltration processes. In order to analyze the variation pattern of the two parameters in the above proposed model, scenarios were set for soil column infiltration in fine-textured soil with a coarse interlayer under different buried depths. A Richards equation-based model (RE-Model) was built for simulating the above scenarios and to obtain the evolution of soil water content along the soil profiles. Simulation results show that the infiltration rate decreased to a constant value when the wetting front crossed the upper interface between the fine and coarse soil layer. The soil matrix suction ($\psi_2$) at the upper interface remained unchanged after the wetting front advanced into the coarse layer, and the steady value of $\psi_2$ showed a linear relationship with the buried depth of the coarse layer. Based on the simulation results of the RE-Model, a method was proposed to determine the saturation coefficients related to the relative hydraulic conductivity and water content at $\psi_2$ in the modified Green-Ampt model. Then, the modified model was tested under various infiltration conditions with different soil layered structures, and the results showed good agreement with the experimental data.

**Keywords:** water infiltration; modified Green-Ampt model; coarse interlayer; unsaturated condition

## 1. Introduction

Soil water infiltration has received considerable interest in agriculture and water research [1–3]. As an important component of the hydrological cycle, infiltration serves as a key role in mitigating flood risk, groundwater contamination control, and supplying water for crop root water uptake in the vadose zone [4,5]. The infiltration can also affect surface runoff in the soil system, which in turn can influence soil erosion [6,7]. Therefore, infiltration research can possibly contribute to land degradation neutrality and restoration of land for the sustainable development of our environment and society [8].

Soil profiles are commonly heterogeneous and have various horizontal layers due to geological processes, formation of crusts, or artificial activities [9]. Horizontal soil layers with different textures show a large variation of hydraulic conductivity and soil water content in the soil profile [10–12]. Therefore, water infiltration behaviors into layered soil are different from that in homogeneous soil in regard to things such as the infiltration rate and the wetting front advancement. For example, the infiltration rate will be reduced to a constant value as the wetting front moves from a fine soil layer to a coarse layer [13], and in this situation, the underlying coarse layer remains unsaturated [14].

Wang et al. [15] found that the constant infiltration rate decreased with the buried depth of sand interlayers in loess soils after the wetting front reached the upper interface of the sand layer.

Infiltration models have been developed to understand the laws embedded in experiments and evaluate the soil water dynamics in different infiltration cases [16,17]. Empirical models, such as the Kostiakov equation (1932), Horton equation (1940), and Philip equation (1957) are widely used because of their simplicity and capability for infiltration computation. However, these models are usually used in homogeneous soils, and may be questionable for simulating water infiltration in layered soils [9]. The Green and Ampt equation (1911) is a half-empirical and half-theoretical model describing the soil water infiltration process, originally in homogeneous soil and later extended to layered soils [18,19]. Wang et al. [13] developed a modified Green-Ampt model for two-layer soil infiltration, and found that the top soil layer's thickness greatly affects layered soil water infiltration. To estimate unstable infiltration in layered soil with non-uniform initial soil water content, Liu et al. [19] derived an infiltration model based on the Green-Ampt method. These modified Green-Ampt models were claimed to reasonably describe infiltration in layered soils. However, the water content and hydraulic conductivity of the wetted zone were usually assumed to be the saturation values in these models, which may not be reasonable for infiltration in fine-textured soil with a coarse interlayer due to the high unsaturation of the coarse interlayer [20].

To account for the unsaturated condition of the infiltrated soil, the effective hydraulic conductivity $K_e$ and the effective soil water content $\theta_e$, were used in some Green-Ampt methods instead of the saturated values [21]. Bouwer [22] suggested the $K_e$ should be half of the saturated hydraulic conductivity. To determine the actual water content and hydraulic conductivity of the wetted zone, Ma et al. [1,23] introduced a saturation coefficient $S_e$ (the ratio of unsaturated soil water content to the saturated one) to the modified Green-Ampt model for layered soils, in which the value of $S_e$ was the same for calculating $K_e$. However, the variation of $K_e$ may not follow the same trend as that of $\theta_e$ during infiltration [24], and using the same $S_e$ for calculating both unsaturated $\theta_e$ and $K_e$ may result in errors. For infiltration in fine-textured soils with a coarse interlayer, the effect of an unsaturated coarse layer on water flow is more apparent, which therefore needs more investigation.

Wang et al. [5] proved that the Richards equation-based numerical simulations were reliable for simulating infiltration in fine-textured soil with a coarse interlayer. Therefore, on the basis of the results obtained from our previous analysis, and in particular, Wang et al. [5], the objectives of this study were to: (a) develop a new modified Green-Ampt model (MGA-2) for water infiltration in fine-textured soil with a coarse interlayer, considering the unsaturated condition of the wetted zone. The testing hypothesis was that the MGA-2 improved the layered model by Wang et al. [13] by accounting for the different unsaturation values behind the wetting front; (b) use a Richards equation-based model to investigate the effect of the coarse interlayer's buried depth on water infiltration, and account for the saturation coefficients for calculating $\theta_e$ and $K_e$ in the MGA-2 based on the simulation results; and (c) test the proposed new model MGA-2 with the infiltration experiments presented in Wang et al., [5] as well as with other two infiltration experiments conducted by the authors.

## 2. The Description of the Modified Green-Ampt Model

We assumed that a soil profile under infiltration with a constant water head consisted of three layers ($j$ = 1, 2, and 3) (see Figure 1, which is a sketch map of the previous lab experiment). The first and third layers were the fine-textured soil, and the coarse layer was in the second layer. The buried depth of the coarse layer meant the depth from the soil's surface to the upper surface of the coarse layer—that is, equal to the thickness of the first layer ($l_1$). The saturated soil water content, the saturated hydraulic conductivity, and the increase of soil water content of the fine-textured soil and coarse soil were $K_{sj}$, $\theta_{sj}$, and $\Delta\theta_{sj}$ ($j$ = 1, 2), respectively. We assumed that each soil layer was homogeneous with a uniform initial water content, and that the soil was evenly wetted by infiltration.

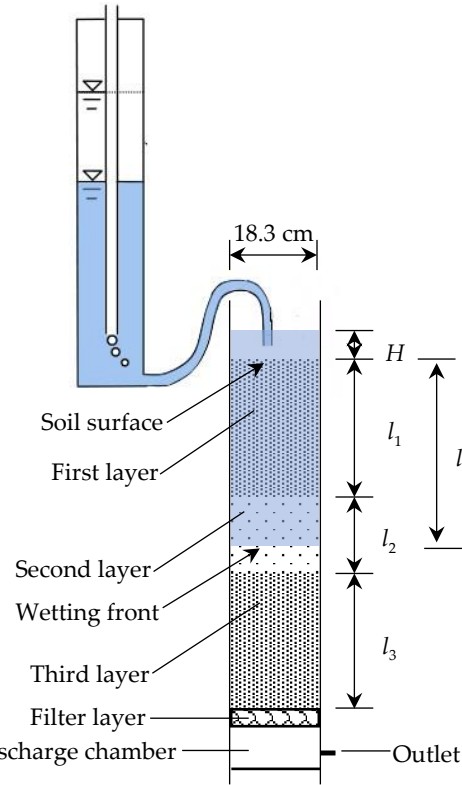

**Figure 1.** Schematic representation of the laboratory infiltration experiment.

When the wetting front was in the first soil layer ($0 < l \leq l_1$), a new modified Green-Ampt model (MGA-2) was used to describe the infiltration process, in which the effective hydraulic conductivity $K_e$ and effective soil water content $\theta_e$ replaced the $K_s$ and $\theta_s$ in the original Green-Ampt model [18]:

$$i = K_{e1} \frac{H + l + \psi_{l1}}{l} \tag{1}$$

$$I = \Delta\theta_1 l \tag{2}$$

$$t = \frac{\Delta\theta_1}{K_{e1}} \left[ l - (\psi_{l1} + H) \ln \frac{H + l + \psi_{l1}}{\psi_{l1} + H} \right] \tag{3}$$

where $i$ (cm min$^{-1}$) and $I$ (cm) are the infiltration rate and cumulative infiltration, respectively, $l$ is the thickness of the wetting zone (cm), $H$ is the ponded water head (cm), $\psi_{l1}$ is the wetting front matrix suction of the first soil layer (cm), and $K_{e1}$ is the effective hydraulic conductivity of the first soil layer (cm min$^{-1}$):

$$K_{e1} = a_1 K_{s1} \tag{4}$$

$$\Delta\theta_1 = b_1 \theta_{s1} - \theta_{01} \tag{5}$$

where $\theta_{01}$ is the initial soil water content of the first soil layer (cm$^3$ cm$^{-3}$), $a_1$ is the saturation coefficient for calculating $K_{e1}$, and $b_1$ is the saturation coefficient for calculating the effective soil water content $\theta_{e1}$.

When the wetting front moves into the sublayer ($l > l_1$), the smaller suction of the coarse soil layer reduces the infiltration rate [25], and the matrix suction at the upper interface of the two soils remains constant after the wetting front reaches the interface [5]. Therefore, by applying Darcy's law to the wetted zone ($0 < l \leq l_1$), the infiltration rate $i$ after the wetting front advances to the coarse layer can be expressed as:

$$i = K'_{e1} \frac{H + l_1 + \psi_2}{l_1} = K'_{e1}\left(1 + \frac{H + \psi_2}{l_1}\right) \tag{6}$$

where $\psi_2$ is the soil matrix suction at the upper interface of the two soils (cm), and $K'_{e1}$ is the effective hydraulic conductivity of the first soil layer when $l > l_1$:

$$K'_{e1} = a_2 K_{s1} \tag{7}$$

where $a_2$ is the saturation coefficient for calculating $K'_{e1}$.

According to Equation (6), for a certain infiltration in a fine-textured soil with a coarse interlayer, $i$ becomes a constant after the wetting front advances to the coarse layer, and the cumulative infiltration $I$ can be expressed as:

$$I = I_1 + (t - t_1) i \tag{8}$$

where $t$ is the arrival time of the wetting front at location $l$ (min), $t_1$ is the wetting front arrival time at the bottom of the first layer (min), and $I_1$ is the cumulative infiltration depth at $t = t_1$ (cm).

The rate of the advancing wetting front is:

$$\frac{dl}{dt} = \frac{i}{\Delta\theta_j}, \ j = 1, 2 \tag{9}$$

Integrating the left side of Equation (9) from 0 to $l$ and the right side from 0 to $t$ gives:

$$l = \begin{cases} l_1 + \frac{i}{\Delta\theta_2}(t - t_1), \ l_1 < l \le (l_1 + l_2) \\ l_1 + l_2 + \frac{i}{\Delta\theta_1}(t - t_2), \ l > (l_1 + l_2) \end{cases} \tag{10}$$

$$\Delta\theta_2 = b_2\theta_{s2} - \theta_{02} \tag{11}$$

where $l_2$ is the thickness of the second layer (cm), and $t_2$ is the arrival time of the wetting front at the bottom of the second layer (min). $\theta_{02}$ is the initial soil water content of the coarse layer (cm$^3$ cm$^{-3}$), and $b_2$ is the saturation coefficient for calculating effective soil water content of the coarse layer.

The water entry suction for homogeneous soil can be estimated by the following method [22,26]:

$$\psi_l = \frac{h_a}{2} \tag{12}$$

$$\frac{\theta - \theta_r}{\theta_s - \theta_r} = \left(\frac{h_a}{|h|}\right)^\lambda = \left(\frac{1}{|\alpha'h|}\right)^\lambda, \ |\alpha'h| > 1 \tag{13}$$

where $\theta$ is the water content (cm$^3$ cm$^{-3}$), $\theta_s$ is the saturated water content (cm$^3$ cm$^{-3}$), $\theta_r$ is the residual water content (cm$^3$ cm$^{-3}$), $h$ is the matrix potential in the unsaturated soil (cm), $h_a$ is the air entry value (cm), $\alpha'$ is an empirical parameter (cm$^{-1}$) and the reciprocal of $h_a$, and $\lambda$ is the pore-size distribution parameter affecting the slope of the retention function.

In the modified Green-Ampt model, the two saturation coefficients $a$, $b$ vary with different soil textures and structures. It is expensive, laborious, and time-consuming to obtain the values of $a$, $b$ through various experiments. A physical-based simulation model (Richards equation) can enhance the insights of the layered soil infiltration process, and the quantification of $a$, $b$ can be analyzed by the simulation results of infiltration in fine-textured soil with different coarse interlayers.

## 3. Materials and Methods

### 3.1. Infiltration Experiment

Two column experiments of infiltration in fine-textured soil with a coarse interlayer are introduced to this study, and the related experimental data are mainly used to testify the proposed modified Green-Ampt model.

### 3.1.1. Experiment I

Experiment I was conducted by Wang et al. [5]. Infiltration was conducted in transparent acrylic columns (18.3 cm inner diameter, 85 cm length) (Figure 1). Four types of soils, a loam (L1) and three sands (S1, S2, S3), were used in this experiment (Table 1). Three groups of soil distribution in the column were designed, that is, loam with a sand interlayer (L1S1L1, L1S2L1, and L1S3L1). The corresponding thicknesses of the three soil layers were 22.5, 20, and 17.5 cm, respectively. The ponded infiltration was conducted with a water head of 2 cm. The infiltration time, infiltration water amount, and wetting front advancement were recorded during the experiment.

**Table 1.** Soil particle size distribution, bulk density, and initial water content for column infiltration experiments.

| Texture | Textural Fractions (%) | | | | Bulk Density (g cm$^{-3}$) | Initial Water Content (cm$^3$ cm$^{-3}$) |
|---|---|---|---|---|---|---|
| | Gravel (>2.0 mm) | Sand (2.0–0.05 mm) | Silt (0.05–0.002 mm) | Clay (<0.002 mm) | | |
| L1 | 0 | 42.2 | 45.8 | 12 | 1.40 | 0.080 |
| S1 | 0.95 | 97.85 | 1.0 | 0.2 | 1.65 | 0.065 |
| S2 | 14.3 | 83.7 | 1.8 | 0.2 | 1.65 | 0.015 |
| S3 | 0 | 96.9 | 2.7 | 0.4 | 1.65 | 0.020 |
| L2 | 0 | 41.9 | 50.4 | 7.7 | 1.50 | 0.014 |
| SL1 | 0 | 55.0 | 39.8 | 5.2 | 1.55 | 0.014 |
| SL2 | 0 | 67.8 | 28.0 | 4.2 | 1.60 | 0.011 |

### 3.1.2. Experiment II

The experiment was conducted in the lab of China Agricultural University. In this experiment, a silt loam (L2) and two sandy loams (SL1, SL2) were used for the layered soil infiltration (Table 1). Two groups of soil distribution in the column were applied, that is, silt loam with a sandy loam interlayer (L2SL1L2, L2SL2L2). The corresponding thickness of the three layers were 22.5, 20, 22.5 cm and 25, 20, 20 cm, respectively, for the two groups. A constant head of 2 cm was applied at the soil's surface. The duration of the infiltration experiment was approximately 600 min. During the experiment, the cumulative infiltration and depth of the wetting front were recorded with time.

### 3.2. The Richards Equation-Based Infiltration Model

In this study, we established a model based on the Richards equation (RE-Model) in order to conduct scenario simulations and to obtain the necessary information for parameter analysis in the new modified Green-Ampt model. The Richards equation is a physical-based model for simulating water dynamics in saturated–unsaturated medium [27]. As stated by Wang et al. [5], although for a distinctly layered soil structure where a fingering flow developed, such a phenomenon cannot be depicted by a one-dimensional Richards equation, Richards equation-based numerical simulations did show good performances of modeling layered soil columns of fine-textured soil with a coarse interlayer. The equation is in the form of:

$$\frac{\partial \theta}{\partial t} = C(h)\frac{\partial h}{\partial t} = \frac{\partial}{\partial z}\left[K(h)\left(\frac{\partial h}{\partial z} - 1\right)\right] \tag{14}$$

where $t$ is time (min), $z$ is the vertical space coordinate in the downward direction from the soil surface (cm), and $C$ is the specific soil water capacity (cm$^{-1}$).

The relationships between $\theta$, $K$, and $h$ were calculated using the van Genuchten-Mualem (VGM) model [28,29]:

$$\theta(h) = \begin{cases} \theta_r + \frac{\theta_s - \theta_r}{(1 + |\alpha h|^n)^{1-1/n}}, & h < 0 \\ \theta_s, & h \geq 0 \end{cases} \tag{15}$$

$$K(h) = K_r K_s = K_s S_e^{0.5} \left[ 1 - \left( 1 - S_e^{n/(n-1)} \right)^{1-1/n} \right]^2 \tag{16}$$

$$S_e = \frac{\theta - \theta_r}{\theta_s - \theta_r} \tag{17}$$

where $\alpha$ is an air-entry parameter (cm$^{-1}$), $n$ is a pore size distribution parameter, and $K_r$ is the relative hydraulic conductivity.

The initial and boundary conditions for ponded infiltration were described by:

$$\theta(z,0) = \theta_0(z), \ 0 < z < Z, \ t = 0 \tag{18}$$

$$h(0,t) = H, \ z = 0, \ t > 0 \tag{19}$$

$$\begin{cases} \frac{\partial(h+z)}{\partial z} = 0, \ h < 0, \ z = Z, \ t > 0 \\ h = 0, \ h \geq 0, \ z = Z, \ t > 0 \end{cases} \tag{20}$$

where $\theta_0$ is the initial soil water content in the soil profile (cm$^3$ cm$^{-3}$), and $Z$ is the maximum length of the simulated soil profile (cm).

The Richards equation was solved with the implicit finite difference method, and the model was compiled in the MATLAB programming language. For numerical simulation, the vertical soil profiles composed of three layers based on the soil textures were used in the above two experiments (Table 1), and further discretized into different compartments (one-dimensional line elements) with uniform spacing of 1 cm. Infiltration rate, cumulative infiltration, and wetting front could be obtained as the simulation results by post-processing.

The RE-Model built in this study was verified by comparison with the widely used HYDRUS-1D model for simulating one-dimensional movement of water, heat, and multiple solutes in variably saturated media [30] based on the infiltration case L1S1L1 in Experiment I. The input VGM parameters of the soils are shown in Table 2. In the later simulation, the verified RE-Mode was used to investigate the infiltration in fine-textured soils with different coarse interlayers.

**Table 2.** The van Genuchten-Mualem (VGM) parameters used in the Richards equation-based model (RE-Model).

| Soil | $\theta_r$ (cm$^3$ cm$^{-3}$) | $\theta_s$ (cm$^3$ cm$^{-3}$) | $\alpha$ (cm$^{-1}$) | $n$ | $K_s$ (cm min$^{-1}$) |
|------|------|------|------|------|------|
| L1 | 0.014 | 0.400 | 0.009 | 1.58 | 0.057 |
| S1 | 0.010 | 0.275 | 0.050 | 2.50 | 0.160 |
| S2 | 0.005 | 0.300 | 0.025 | 2.50 | 0.070 |
| S3 | 0.005 | 0.300 | 0.018 | 4.30 | 0.194 |
| L2 | 0.010 | 0.420 | 0.012 | 1.40 | 0.021 |
| SL1 | 0.008 | 0.330 | 0.025 | 2.30 | 0.027 |
| SL2 | 0.005 | 0.310 | 0.046 | 2.20 | 0.031 |

### 3.3. Scenario Analysis

To investigate the effects of buried depth of the coarse interlayer on water infiltration with the RE-Model, five different buried depths (20, 22.5, 30, 40, and 50 cm) of sand layers (20 cm thick) embedded in the loam were set based on the soil profiles (L1S1L1, L1S2L1, and L1S3L1) in Experiment I. For model inputs in the scenario simulations, the initial conditions were kept the same with the experiments except for the locations of the coarse interlayer, and the VGM parameters of L1, S1, S2, and S3 verified by Wang et al. [5] are listed in Table 2. The simulated infiltration rate, cumulative infiltration, water content, and hydraulic conductivity alongside the soil profile were analyzed, and the hydraulic parameters used in the MGA-2 were determined.

In the scenario analysis, the simulation soil profile extended from the soil surface to a depth of 150 cm, and the simulation time was set to 300 min.

### 3.4. The Test of the Proposed Green-Ampt Model

The proposed MGA-2 was used to describe the infiltration processes in fine-textured soils with a coarse interlayer based on the infiltration Experiments I and II, and the saturation coefficients in MGA-2 were determined by the methods deduced in the scenario analysis. The infiltration cases were also calculated by a modified Green-Ampt model (MGA-1) for layered soil infiltration [13]. The main difference of MGA-1 from MGA-2 was that it assumed the soil water content and hydraulic conductivity behind the wetting front were equal to the saturation values.

To evaluate the new modified Green-Ampt model, the root mean square error (RMSE) was used as a criterion to reflect the goodness of calculation, which can be expressed as:

$$\text{RMSE} = \sqrt{\frac{1}{N} \sum_{i=1}^{N} (O_i - P_i)} \tag{21}$$

where $O$ is the observed value, $P$ is the calculated value, and $N$ is the number of observations.

## 4. Results and Discussion

### 4.1. The Verification of the RE-Model

A comparison was conducted for the RE-Model and the HYDRUS-1D model based on the infiltration case L1S1L1. As shown in Figure 2, the simulated soil water content profiles are almost the same at the infiltration times of 30 and 150 min by the two models, which indicates the reliability of the RE-Model.

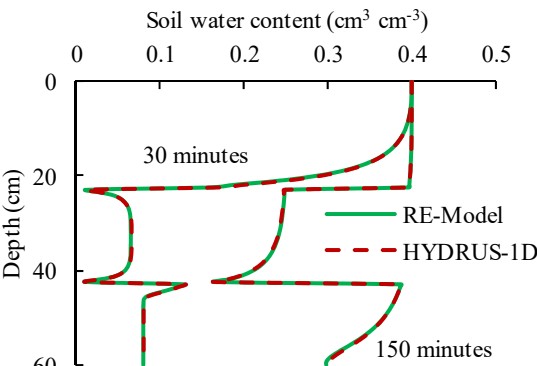

**Figure 2.** The comparison of the simulated soil water content profiles from the RE-model and HYDRUS-1D for the infiltration case L1S1L1.

### 4.2. The Effect of Buried Depth of Coarse Interlayer on Water Infiltration

The infiltrations in a loamy soil with different buried depths of sand interlayer were simulated by the RE-Model. The variations of the infiltration rate and cumulative infiltration with time under different buried depths of sand interlayer (case L1S1L1) are shown in Figure 3. When the water moves in the upper loam of the layered soil, the infiltration process has no difference compared with that in homogeneous loam. Once the water enters the sand interlayer, the infiltration rate reduces to a constant value (called the steady infiltration rate), and the cumulative infiltration increase down and turns to vary linearly with time. By using the column experiment and numerical simulation, Wang et al. [5] found the same phenomenon in infiltration into fine-textured soils with a sand interlayer. The steady infiltration rate becomes smaller with an increase of the sand layer's buried depth. The shallow buried depth of the coarse layer has a greater impact on the steady infiltration rate and cumulative infiltration compared with the infiltration results in homogeneous soil.

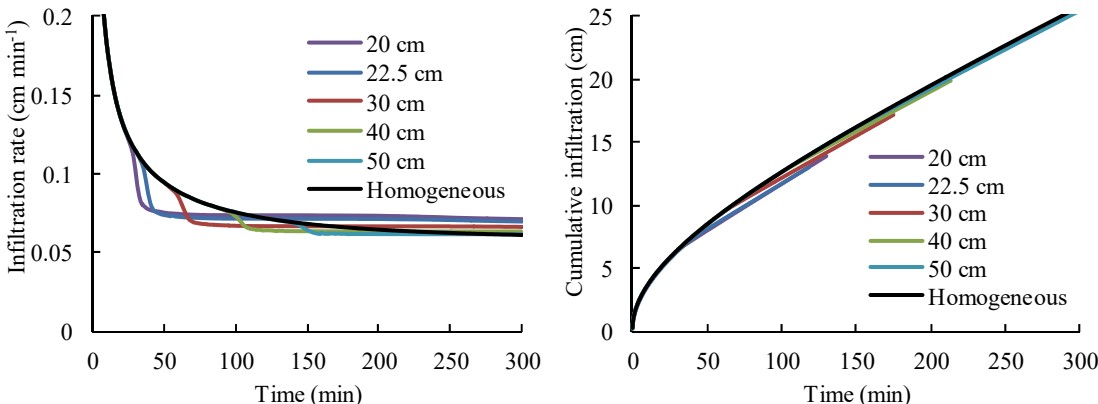

**Figure 3.** The variation of infiltration rate (**left**) and cumulative infiltration (**right**) with time under different buried depths of the sand interlayer for case L1S1L1.

### 4.3. Parameter Determination for the MGA-2

The simulated soil matrix potential profiles for L1S1L1 with a sand buried depth of 20 cm are shown in Figure 4. The soil matrix potential at the upper interface $(-\psi_2)$ of the two soils remains unchanged after the wetting front advances into the sand layer ($t > 40$ min). According to Equation (6), the infiltration rate in the sublayer turned to a constant value, which confirms the proposed model. The steady soil matrix suction $\psi_2$ shows a linear relationship with the buried depth of the sand layer, and its values rise by about 1 cm for every 10 cm increase in buried depth of sand layer for infiltration cases in this scenario (Figure 5), which may be caused by the larger hydraulic loss in the first layer for a deep buried depth of the coarse interlayer. The water entry suction $\psi_l$ for homogeneous soils (S1, S2, and S3) calculated by Equation (12) is close to the $\psi$ when the buried depth of the sand layer is equal to 0 in Figure 5 (Table 3). Therefore, the relationship of the steady soil matrix suction $\psi_2$ and the buried depth of coarse layer can be estimated by:

$$\psi_2 = \frac{h_a}{2} + \eta l_1 \tag{22}$$

where $\eta$ is a coefficient to determine the increment of $\psi_2$ with $l_1$.

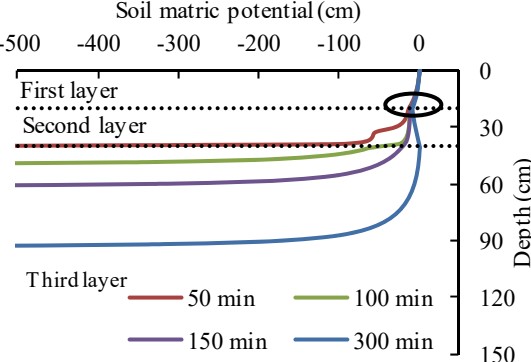

**Figure 4.** The simulated soil matrix potential profiles for L1S1L1 at different infiltration times.

**Table 3.** The water entry suction for homogeneous soils in Equation (12) and in Figure 5 when the buried depth of sand layer is equal to 0, and parameter $\eta$ for the two infiltration experiments.

| Soil | $\psi_l$ in Equation (12) | $\psi$ in Figure 5 | $\eta$ |
|------|---------------------------|--------------------|--------|
| S1 | 8.4 | 7.9 | |
| S2 | 10.5 | 10.1 | 0.1 |
| S3 | 41.0 | 42.6 | |
| SL1 | 10.7 | 10.2 | |
| SL2 | 3.0 | 2.4 | 0.14 |

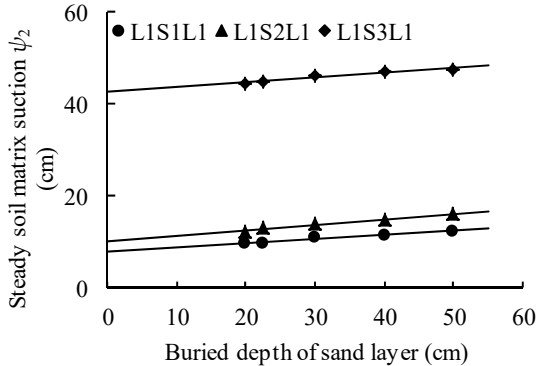

**Figure 5.** The variation of the simulated soil matrix suction at the upper interface of the two soils with the buried depth of sand layer for cases L1S1L1, L1S2L1, and L1S3L1 during the steady infiltration stage.

The simulated soil profiles of hydraulic conductivity and water content after infiltration were analyzed to study the soil unsaturated condition in fine-textured soil with a coarse interlayer. Here, we used the simulation results of a case with the sand interlayer buried at a depth of 50 cm as an example. Figures 6 and 7 show the distribution of relative hydraulic conductivity and soil water content along the profile after an infiltration time of 300 min, respectively. Note that the soil water content in S3 differs from the other two in the sand interlayer—that is, the soil water content increases with depth in this sand interlayer, while the other two have a decreasing trend (as shown in Figure 7). This is because of the difference in hydraulic property for the three-sand interlayer—for example, S3 has a much higher water entry suction (45.0 cm) than S1 and S2 (9.9 cm and 12.8 cm), as shown in Table 4. Normally, the matrix potential is in a continuous manner in layered soil, while the soil water content may have a great difference at the interface because of the difference in soil hydraulic property.

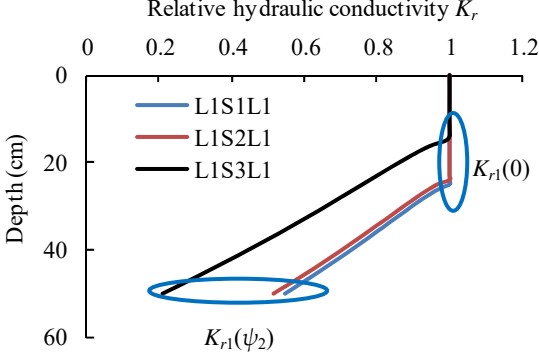

**Figure 6.** The distribution of relative hydraulic conductivity in the first layer when the buried depth of sand is 50 cm and the infiltration time is 300 min.

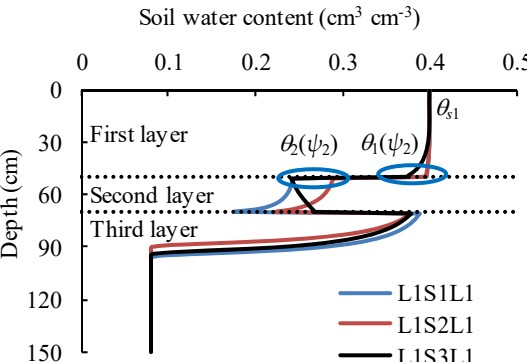

**Figure 7.** The distribution of soil water content along the soil profile when the buried depth of sand is 50 cm and the infiltration time is 300 min.

**Table 4.** Parameters of the new, modified Green-Ampt model for the laboratory infiltration experiment.

| Cases | $a_1$ | $b_1$ | $a_2$ | $b_2$ | $\psi_2$ | $\psi_{l1}$ | $\theta_1(\psi_2)$ | $\theta_2(\psi_2)$ | $K_{r1}(\psi_2)$ |
|-------|-------|-------|-------|-------|----------|-------------|--------------------|--------------------|------------------|
| L1S1L1 | 0.953 | 1.000 | 0.907 | 0.911 | 9.9 | | 0.397 | 0.251 | 0.568 |
| L1S2L1 | 0.941 | 1.000 | 0.881 | 0.968 | 12.8 | 30.4 | 0.395 | 0.290 | 0.513 |
| L1S3L1 | 0.839 | 0.997 | 0.678 | 0.775 | 45.0 | | 0.371 | 0.232 | 0.197 |
| L2SL1L2 | 0.873 | 1.000 | 0.747 | 0.963 | 12.6 | 12.0 | 0.412 | 0.318 | 0.288 |
| L2SL2L2 | 0.925 | 1.000 | 0.849 | 0.935 | 5.1 | | 0.418 | 0.290 | 0.451 |

The distribution of relative hydraulic conductivity $K_{r1}$ in the first layer changed a little after the wetting front advanced into the sand interlayer, and the value of $K_{r1}$ can reflect the unsaturation condition of the hydraulic conductivity in the wetted zone. According to the proportion of $K_{r1}$ in the first layer (Figure 6), the saturation coefficient $a_2$ for calculating the effective hydraulic conductivity $K'_{e1}$ is approximately equal to:

$$a_2 = 1 - \frac{\frac{1}{2}\frac{K_{r1}(0)-K_{r1}(\psi_2)}{K_{r1}(0)}l_1[K_{r1}(0) - K_{r1}(\psi_2)]}{l_1 K_{r1}(0)} = 1 - \frac{[1 - K_{r1}(\psi_2)]^2}{2} \tag{23}$$

When soil water flows in the first layer, the hydraulic conductivity behind the wetting front is firstly almost equal to the saturation value, and then decreases due to the unsaturation at the bottom of the first layer (Figure 6). Therefore, the saturation coefficient $a_1$ for calculating the effective hydraulic conductivity $K_{e1}$ when $l < l_1$ is approximately expressed as:

$$a_1 = \frac{1 + a_2}{2} \tag{24}$$

Based on the above analysis, the saturation coefficient $b_1$ for calculating the effective water content of the loam can be estimated by (Figure 7):

$$b_1 = 1 - \frac{\frac{1}{2}\frac{\theta_{s1}-\theta_1(\psi_2)}{\theta_{s1}}l_1[\theta_{s1} - \theta_1(\psi_2)]}{l_1 \theta_{s1}} = 1 - \frac{1}{2}\left[\frac{\theta_{s1} - \theta_1(\psi_2)}{\theta_{s1}}\right]^2 \tag{25}$$

The averaged soil water content of the sand interlayer is an approximation to the $\theta(\psi_2)$ when water advances into this layer. Therefore, the saturation coefficient $b_2$ for calculating the effective water content of the sand interlayer can be expressed as:

$$b_2 = \frac{\theta_2(\psi_2)}{\theta_{s2}} \tag{26}$$

The saturation coefficients calculated through the proposed methods were compared with the simulation results by the RE-Model. As shown in Table 5, the estimated values of saturation coefficients with Equations (23)–(26) are very close to the simulation analysis results, except for a few cases. This result indicates that the methods for estimating the saturation coefficient values are reasonable for infiltration in fine-textured soil with a coarse interlayer.

**Table 5.** Comparison of the saturation coefficients simulated by the RE-Model and calculated by the modified Green Ampt model (MGA-2) with different buried depths of sand for cases L1S1L1, L1S2L1, and L1S3L1.

| Case | Buried Depth of Sand (cm) | RE-Model | | | MGA-2 | | |
|------|---------------------------|----------|----------|----------|----------|----------|----------|
| | | $a_2$ | $b_1$ | $b_2$ | $a_2$ | $b_1$ | $b_2$ |
| L1S1L1 | 20 | 0.845 | 0.999 | 0.944 | 0.910 | 1.000 | 0.918 |
| | 30 | 0.852 | 0.999 | 0.900 | 0.898 | 1.000 | 0.892 |
| | 40 | 0.866 | 0.999 | 0.854 | 0.893 | 1.000 | 0.879 |
| | 50 | 0.879 | 0.999 | 0.821 | 0.884 | 1.000 | 0.858 |
| L1S2L1 | 20 | 0.894 | 0.999 | 0.990 | 0.884 | 1.000 | 0.970 |
| | 30 | 0.848 | 0.998 | 0.976 | 0.873 | 1.000 | 0.961 |
| | 40 | 0.841 | 0.998 | 0.933 | 0.864 | 1.000 | 0.954 |
| | 50 | 0.849 | 0.998 | 0.893 | 0.854 | 1.000 | 0.945 |
| L1S3L1 | 20 | 0.715 | 0.991 | 0.841 | 0.680 | 0.997 | 0.784 |
| | 30 | 0.680 | 0.989 | 0.840 | 0.673 | 0.997 | 0.756 |
| | 40 | 0.684 | 0.988 | 0.746 | 0.669 | 0.997 | 0.741 |
| | 50 | 0.708 | 0.989 | 0.665 | 0.667 | 0.997 | 0.732 |

*4.4. The Performance of the Proposed Green-Ampt Model*

The layered soil infiltration experiments, that is, experiments I and II, were used to test the new modified Green-Ampt model (MGA-2). The saturation coefficients in the MGA-2 can be calculated by using Equations (23)–(26) and parameter values in Table 3, and their corresponding values are listed in Table 4. The calculated infiltration rate, cumulative infiltration, and wetting front advancement of the two experiments are shown in Figures 8–10 and Figures 11–13, respectively. The calculated results by MGA-1 are also shown in Figures 8–13.

As shown in Figures 8–10 and Figures 11–13, the MGA-1 and MGA-2 have little difference in simulating infiltration rate, cumulative infiltration, and wetting front advancement in the upper loam, and both can capture the observed values. This is because the effect of unsaturation in the upper loam is small after infiltration. When soil water advances into the sand interlayer, the larger pores in the coarse sand and the lower infiltration rate than the potential can lead to a high level of unsaturation in the sand and surrounding soils [31], which would have a great effect on water infiltration. Therefore, the MGA-1 overestimated the water infiltration processes in this stage. For case L1S3L1, the calculated infiltration rate, cumulative infiltration, and wetting front advancement by MGA-1 are much higher than the observations. This is because the steady soil matrix suction of the sand interlayer ($\psi_2$) is much higher (Table 5), and the soil unsaturated condition is more serious. However, the MGA-2 shows a relatively good agreement with the observed data (Figures 8–13), which can be verified by the smaller RMSE values of the calculated infiltration rate, cumulative infiltration, and wetting front (Table 6).

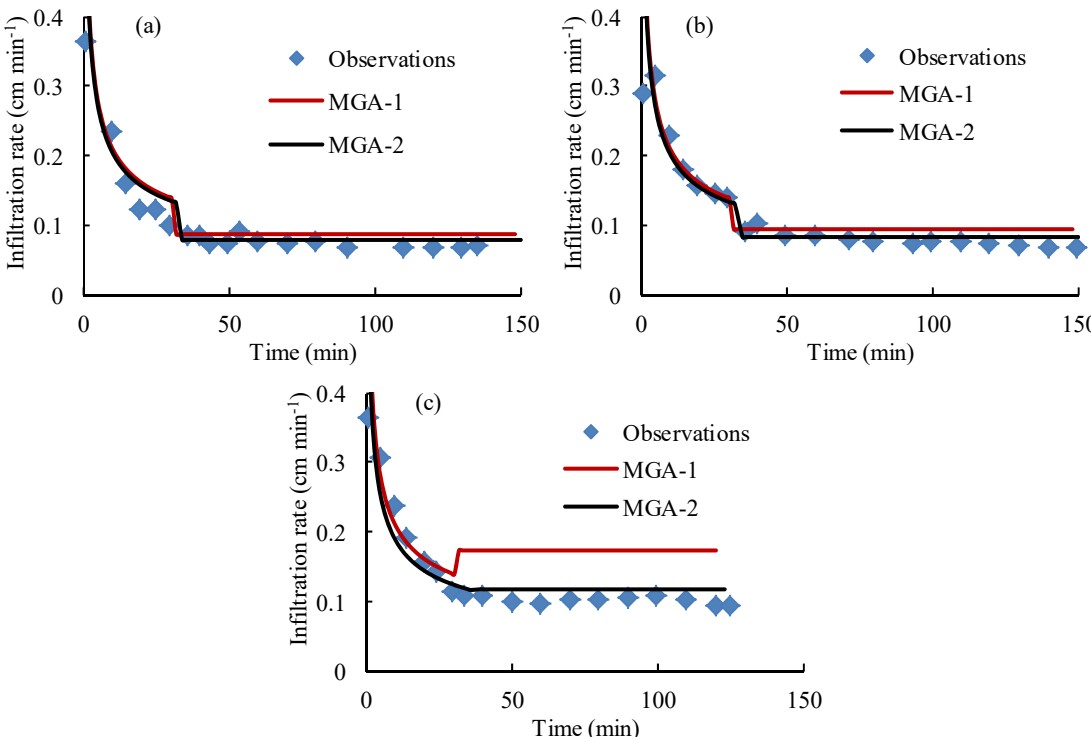

**Figure 8.** Comparison of the observed infiltration rate and the calculated value with the modified Green-Ampt models for cases (**a**) L1S1L1, (**b**) L1S2L1, and (**c**) L1S3L1 in Experiment I.

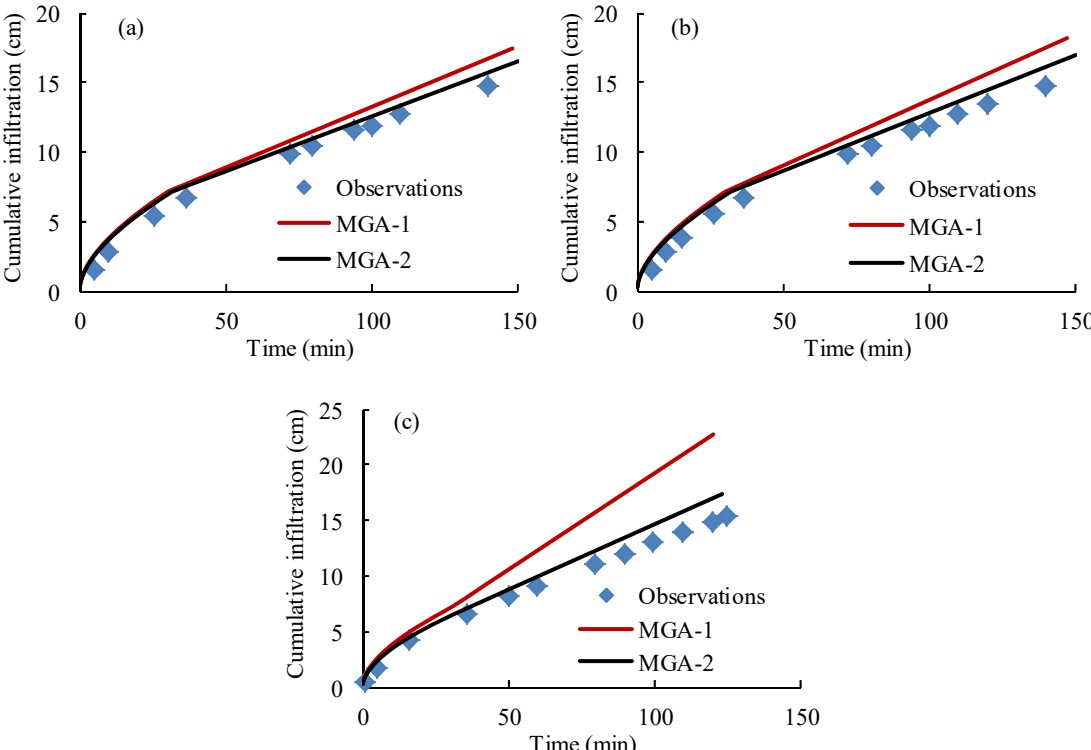

**Figure 9.** Comparison of the observed cumulative infiltration and the calculated value with the modified Green-Ampt models for cases (**a**) L1S1L1, (**b**) L1S2L1, and (**c**) L1S3L1 in Experiment I.

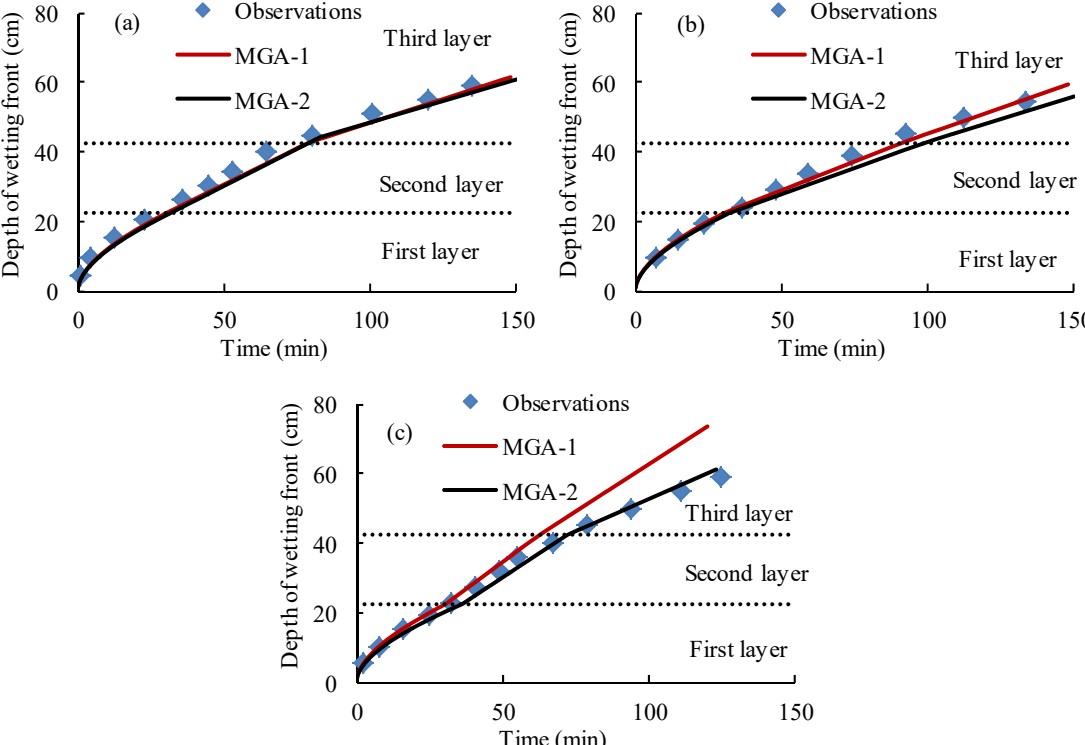

**Figure 10.** Comparison of the observed wetting front advancement and the calculated value with the modified Green-Ampt models for cases (**a**) L1S1L1, (**b**) L1S2L1, and (**c**) L1S3L1 in Experiment I.

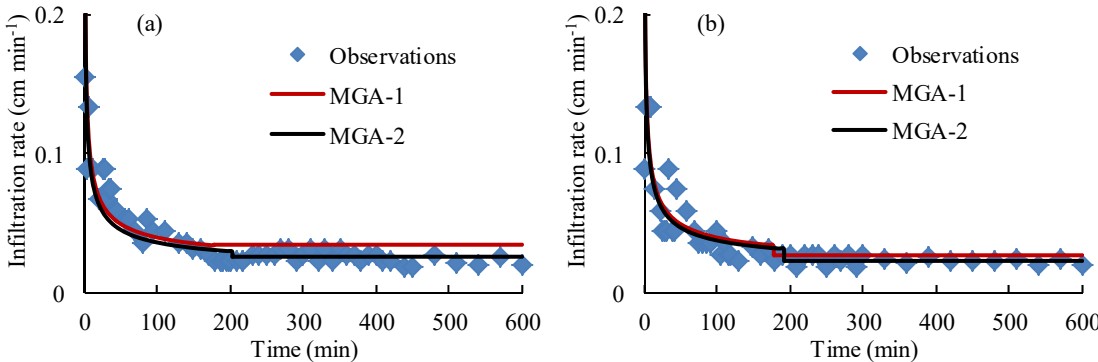

**Figure 11.** Comparison of the observed infiltration rate and the calculated value with the modified Green-Ampt models for cases (**a**) L2SL1L2 and (**b**) L2SL2L2 in Experiment II.

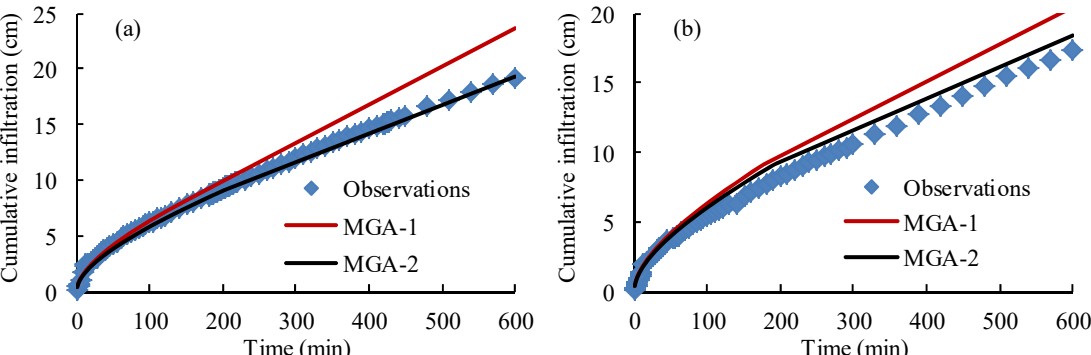

**Figure 12.** Comparison of the observed cumulative infiltration and the calculated value with the modified Green-Ampt models for cases (**a**) L2SL1L2 and (**b**) L2SL2L2 in Experiment II.

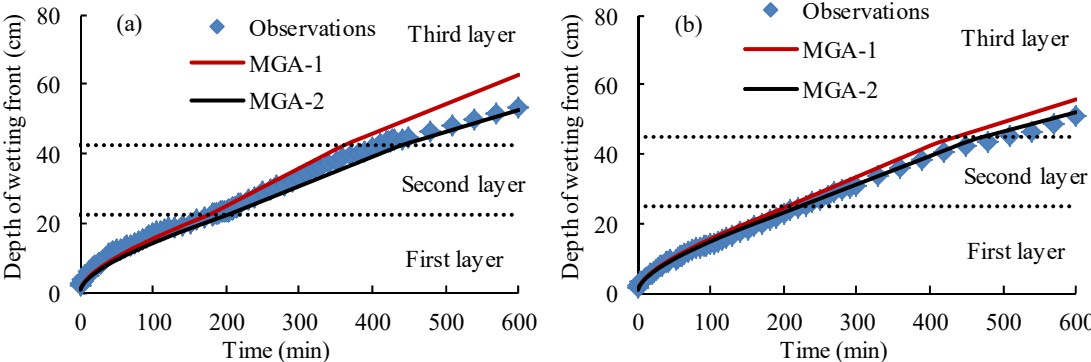

**Figure 13.** Comparison of the observed wetting front advancement and the calculated value with the modified Green-Ampt models for cases (**a**) L2SL1L2 and (**b**) L2SL2L2 in Experiment II.

**Table 6.** The root mean square error (RMSE) values for the simulation results of the MGA-1 and MGA-2.

| Items | Cases | MGA-1 | MGA-2 |
|---|---|---|---|
| Infiltration rate (cm min$^{-1}$) | L1S1L1 | 0.02 | 0.01 |
| | L1S2L1 | 0.02 | 0.01 |
| | L1S3L1 | 0.06 | 0.02 |
| | L2SL1L2 | 0.01 | 0.01 |
| | L2SL2L2 | 0.01 | 0.01 |
| Cumulative infiltration (cm) | L1S1L1 | 1.26 | 0.76 |
| | L1S2L1 | 1.66 | 0.83 |
| | L1S3L1 | 4.72 | 1.26 |
| | L2SL1L2 | 1.81 | 0.35 |
| | L2SL2L2 | 1.91 | 0.87 |
| Wetting front depth (cm) | L1S1L1 | 2.00 | 2.09 |
| | L1S2L1 | 1.29 | 3.22 |
| | L1S3L1 | 5.20 | 1.51 |
| | L2SL1L2 | 3.65 | 2.18 |
| | L2SL2L2 | 2.80 | 0.43 |

The MGA-2 assumes that a sharp wetting front separates the soil profile into an upper saturated zone and a lower unsaturated zone, which is a simplification of the actual infiltration condition. In order to find out whether the assumption is reasonable or not, a comparison was conducted between the time $t_1$ after the wetting front reached $l$ for the RE-Model simulation and the infiltration time $t_2$ after the cumulative infiltration reached the value of $I$ at $l$ for the MGA-2 calculation (Table 7). For more permeable soil (L1S1L1, L1S2L1, and L1S3L1), the times $t_1$ and $t_2$ differ slightly, which indicates that a nearly piston-shaped wetting front is suitable for these soils. For less permeable soil (L1SL1L1, L1SL2L1), the time $t_1$ differs greatly from $t_2$ due to larger water storage in the wetting zone. Therefore, the MGA-2 model is more feasible for infiltration in permeable soil.

**Table 7.** The time $t_1$ after the wetting front reached $l$ for the RE-Model simulation, and the infiltration time $t_2$ after the cumulative infiltration reached the value of $I$ at $l$ for the MGA-2 calculation.

| The Depth of Wetting Front $l$ (cm) | L1S1L1 | | L1S2L1 | | L1S3L1 | | L2SL1L2 | | L2SL2L2 | |
|---|---|---|---|---|---|---|---|---|---|---|
| | $t_1$ | $t_2$ | $t_1$ | $t_2$ | $t_1$ | $t_2$ | $t_1$ | $t_2$ | $t_1$ | $t_2$ |
| 10 | 5.71 | 7.56 | 5.71 | 7.67 | 5.71 | 8.55 | 33.06 | 54.74 | 33.06 | 51.73 |
| 30 | 49.12 | 49.47 | 53.36 | 57.12 | 47.23 | 49.58 | 260.00 | 284.10 | 265.99 | 276.60 |
| 50 | 100.97 | 106.08 | 121.92 | 127.06 | 95.00 | 92.46 | 526.66 | 548.70 | 500.00 | 552.10 |

## 5. Discussion

The previously discussed results show that the proposed MGA-2 model is reasonable to describe the water infiltration in fine-textured soil with a coarse interlayer. Compared with the Green-Ampt model (1911) and the MGA-1, the MGA-2 has two advantages. Firstly, the MGA-2 can calculate infiltration processes in fine-textured soil with a coarse interlayer by considering the soil unsaturated conditions. Secondly, the saturation coefficients for calculating the effective soil hydraulic conductivity and soil water content can simply be determined from the soil's hydraulic properties.

In addition to simulating infiltration into fine-textured soil with a coarse interlayer, the proposed MGA-2 can be extended to simulate infiltration in some other soil cases, such as: (1) a deeply buried straw layer in the soil serving as a water flow barrier to increase the amount of stored soil water available for plant use [32]; and (2) a compacted shallow soil layer, and possibly a permeable deeper soil layer caused by heavy machinery or drainage [8]. The soil-straw-soil or compacted-loose soil system is similar to the fine-textured soil with a coarse interlayer. Therefore, the MGA-2 can be potentially applied under these circumstances to assess the soil water-flow process and to better manage crop production and soil restoration.

## 6. Conclusions

A new modified Green-Ampt model was proposed to predict the infiltration process in fine-textured soil with a coarse interlayer, considering the effect of unsaturation. In the proposed model, two saturation coefficients were introduced to calculate the effective soil water content ($\theta_e$) and hydraulic conductivity ($K_e$). To study the effect of the buried depth of the coarse interlayer on water infiltration, and to estimate the soil hydraulic parameters in the new modified Green-Ampt model, a physical-based model was built based on the Richards equation. Then, based on the infiltration experiment conducted by Wang et al. [5], an assessment of the infiltration in fine-textured soil with a coarse interlayer under various buried depths was carried out by the Richards equation-based model.

The simulation results by the Richards equation indicate that the sand interlayer in fine-textured soils could reduce the infiltration rate to a constant value and turn cumulative infiltration to a linear variation with time. The steady infiltration rate becomes smaller with the increase of the sand layer's buried depth. The soil matrix suction at the upper interface ($\psi_2$) of the two soils remains unchanged after the wetting front advances into the coarse layer, and the value of $\psi_2$ shows a linear relationship with the buried depth of the coarse layer. The saturation coefficients ($a_1$ and $a_2$) for calculating $K_e$ can be determined by the relative hydraulic conductivity $K_{r1}(\psi_2)$ of the fine-textured soil, and the saturation coefficients ($b_1$ and $b_2$) for calculating $\theta_e$ can be determined by the $\theta_s$ and $\theta(\psi_2)$ of the two soils.

The proposed modified Green-Ampt model and methods for calculating the soil hydraulic parameters were tested based on the infiltration experiments presented by Wang et al. [5] and conducted by the authors. Good agreement between the measured infiltration data and the calculating values by the proposed theory indicates that the proposed MGA-2 model can reasonably simulate infiltration in fine-textured soil with a coarse interlayer.

**Author Contributions:** X.M. conceived the research theme; C.W. provided data; S.C. performed analysis and wrote the paper.

**Funding:** This work was supported by the National Natural Science Foundation of China (51790535, 51379207).

**Acknowledgments:** We give special thanks to the anonymous editors and reviewers for their valuable comments, which improved this manuscript's quality.

**Conflicts of Interest:** The authors declare no conflict of interest.

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
