# Peer review of "A Modified Green-Ampt Model and Parameter Determination for Water Infiltration in Fine-textured Soil with Coarse Interlayer"

_water, doi:10.3390/w11040787_

Reviewer 1 Report

Line 176: What is the thickness of the sand layer?
Line 202: Instead of "were shown" it should be "are shown".
Line 215, 216: Instead of "were shown" it should be "are shown".
Figure 3: Add the location of the three layers to the graph.
Figure 6: This figure needs more explanation: The course of the black line (L1S3L1) differs from the other two for depth greater than 50 cm. The soil water content increases with depth, this makes no sense to me.
Figure 6 (continued): The infiltration has already reached the third layer, so the lines should be continued to the third third layer as well. This could also help the reader to understand the course of the L1S3L1 line (see above).
Line 284 to 287: If you would like to add an estimation on the shape of the infiltration, you could follow Sinaba, B., Becker, B., Klauder, W., Salazar, I. and Schüttrumpf, H. (2013): On the proceeding of a saturation front under ponded conditions. Obras y Proyectos 13, 31-39.  In the section "Transition zone" The authors compare the time to reach cumulative infiltrated value that corresponds to an infiltration depth of 1 m and the time to reach a wetting front of 1 m and conclude a "nearly piston-shaped wetting front of the saturation front [...] for the sand soil". A quite straight-forward method. You have the advantage that you did simulations with the RE model (which is not the case for Sinaba et al. 2013). I believe that Figure 3 of your paper gives some hints on the shape of the wetting front as well.

Author Response

1. Line 176: What is the thickness of the sand layer?

Added. The thickness of the sand layer is 20 cm for the soil profiles (L1S1L1, L1S2L1 and L1S3L1) in the scenario analysis. See Line 181.

2. Line 202: Instead of "were shown" it should be "are shown".

Corrected. See Line 205.

3. Line 215, 216: Instead of "were shown" it should be "are shown".

Corrected. See Line 218-219.

4. Figure 3: Add the location of the three layers to the graph.

Added. See Fig. 3 (Line 231).

5. Figure 6: This figure needs more explanation: The course of the black line (L1S3L1) differs from the other two for depth greater than 50 cm. The soil water content increases with depth, this makes no sense to me.

The difference of L1S3L1 from the L1S1L1 and L1S2L1 is caused by hydraulic property for the three sand interlayer, e.g., S3 has a much higher water entry suction (i.e., 45.0 cm) than S1 and S2 (9.9 cm and 12.8 cm). Therefore, the soil water content of S3 is more easily reach the steady state.

For the increase of soil water content with depth in case L1S3L1, this can be attributed to the water entry suction of the loam in the 3rd layer (L1) and sand (S1, S2, S3). The water entry suction of loam L1 (30.4 cm) is lower than that of S3 (45 cm), but larger than S1 and S2 (9.9 cm and 12.8 cm). Normally the matrix potential is in a continuous manner in layered soil (See Figure 1 in this cover letter). Therefore, the variation of water matrix potential of S3 shows a increasing trend with depth at this infiltration time (t=300 min).

We have added the relevant explanations in the revised manuscript. Line 241-247.

Figure 1. The distribution of soil matrix potential along the soil profile when the buried depth of sand is 50 cm and the infiltration time is 300 minutes.

6. Figure 6 (continued): The infiltration has already reached the third layer, so the lines should be continued to the third third layer as well. This could also help the reader to understand the course of the L1S3L1 line (see above).

Modified. See Fig. 6.

7. Line 284 to 287: If you would like to add an estimation on the shape of the infiltration, you could follow Sinaba, B., Becker, B., Klauder, W., Salazar, I. and Schüttrumpf, H. (2013): On the proceeding of a saturation front under ponded conditions. Obras y Proyectos 13, 31-39.  In the section "Transition zone" The authors compare the time to reach cumulative infiltrated value that corresponds to an infiltration depth of 1 m and the time to reach a wetting front of 1 m and conclude a "nearly piston-shaped wetting front of the saturation front [...] for the sand soil". A quite straight-forward method. You have the advantage that you did simulations with the RE model (which is not the case for Sinaba et al. 2013). I believe that Figure 3 of your paper gives some hints on the shape of the wetting front as well.

Thank you very much for this advice.

In the revised manuscript, in order to find out whether the assumption (a sharp wetting front separates the soil profile into an upper saturated zone and a lower unsaturated zone) is reasonable or not for MGA-2, a comparison has been conducted between the time t1 after the wetting front reached l for the RE-Model simulation and the infiltration time t2 after the cumulative infiltration reached the value of I at l for the MGA-2 calculation. See Line 295-303 and Table 7.

Reviewer 2 Report

Review Water 469144

The authors are resubmitting a previously-rejected manuscript that I also reviewed. In my opinion, this new paper faces a new potential rejection.The key aspect is that (in my opinion) the paper should clarify much more what is novel and what is not in this new work. Right now, it presents fundamentally the same key problem as in the previous submission: it confuses the reader as it does not acknowledge that part of the results are obtained from previous works, particularly Wang et al 2014.

In the cover letter, the authors indicate that this paper (quoting) “is a further research based on our previous study (Wang et al., 2014).”, which I think is definitely OK. However, I don’t think that the problem is  that the authors “had added too many similar narrative (regarding the experiment)” to their previous research. Basically, they used the SAME experimental results obtained in Wang et al 2014 in this new work (including both column data and RE-model), and improperly claimed in the previous paper that the results are new. The authors seem recalcitrant in this sense: see for instance the objective (d) of this new paper (“test the proposed Green-Ampt model with the infiltration experiments conducted in this study”.) These infiltration experiments were not conducted in this study!

I don’t understand what the problem is to acknowledge that the authors have revisited some old data with a new model analysis! It would be very easy to use the same words adopted in the cover letter:

·        “This manuscript is a further research based on our previous study (Wang et al., 2014).”

·        “Wang et al. (2014) proved that the Richards equation-based numerical simulations were reliable to simulate infiltration into fine-textured soil with a coarse interlayer. Therefore, we applied this conclusion to our study, and the RE-Mode was directly used to investigate the cases of infiltration in fine-textured soil with different coarse interlayers in this study. “

·        In the application of the modified Green-Ampt model (MGA-2), the MGA-2 was not only tested by the experimental data from Wang et al. (2014), but also tested by the other measured data conducted by the authors.

These three statements are very clear and allow the reader understanding what has been done before. They could be put for instance towards the end of the introduction. Right after these sentences (as well as others that the Authors consider adequate to properly introduce previous results to the reader), the “Objective” of this paper can be now stated.  Quoting from the new submission, I suggest for instance something like:

“On the basis of the results obtained from our previous analysis, and in particular, Wang et al (1999, 2014), the objectives of this study are to:

(a) develop a NEW modified Green-Ampt model for water infiltration in fine textured soil with coarse interlayer, considering the unsaturated condition of the wetted zone; THE TESTING HYPOTHESIS IS THAT NEW MODEL MGA-2 IMPROVES THE LAYERED MODEL BY WANG ET AL (1999, called here after MGA-1) BY ACCOUNTING FOR THE DIFFERENT SATURATION VALUES BEHIND THE WETTING FRONT.

(b) use a Richards equation-based model and experimental results presented in WANG et al 2014 to investigate the effect of coarse interlayer’s buried depth on ware infiltration using THE NEW MGA-2 MODEL, that accounts for the saturation coefficients for calculating e and Ke based on the simulation results;

(c) test the proposed new model MGA-2 with the infiltration experiments and Richards model  PRESENTED IN WANG ET AL 2014 (and not in this study!) to verify mga-2.

Note that what I wrote above is just an example of possible changes, and the authors are obviously free to adapt the text as they best prefer. The point is that the manuscript must be much clearer in this sense, and similar changes should be applied elsewhere in the manuscript, including the conclusions and the abstract.

Author Response

The authors are resubmitting a previously-rejected manuscript that I also reviewed. In my opinion, this new paper faces a new potential rejection.The key aspect is that (in my opinion) the paper should clarify much more what is novel and what is not in this new work. Right now, it presents fundamentally the same key problem as in the previous submission: it confuses the reader as it does not acknowledge that part of the results are obtained from previous works, particularly Wang et al 2014.

In the cover letter, the authors indicate that this paper (quoting) “is a further research based on our previous study (Wang et al., 2014)”, which I think is definitely OK. However, I don’t think that the problem is that the authors “had added too many similar narrative (regarding the experiment)” to their previous research. Basically, they used the SAME experimental results obtained in Wang et al 2014 in this new work (including both column data and RE-model), and improperly claimed in the previous paper that the results are new. The authors seem recalcitrant in this sense: see for instance the objective (d) of this new paper (“test the proposed Green-Ampt model with the infiltration experiments conducted in this study”.) These infiltration experiments were not conducted in this study!

I don’t understand what the problem is to acknowledge that the authors have revisited some old data with a new model analysis! It would be very easy to use the same words adopted in the cover letter:

·        “This manuscript is a further research based on our previous study (Wang et al., 2014).”

·        “Wang et al. (2014) proved that the Richards equation-based numerical simulations were reliable to simulate infiltration into fine-textured soil with a coarse interlayer. Therefore, we applied this conclusion to our study, and the RE-Mode was directly used to investigate the cases of infiltration in fine-textured soil with different coarse interlayers in this study. “

·        In the application of the modified Green-Ampt model (MGA-2), the MGA-2 was not only tested by the experimental data from Wang et al. (2014), but also tested by the other measured data conducted by the authors.

These three statements are very clear and allow the reader understanding what has been done before. They could be put for instance towards the end of the introduction. Right after these sentences (as well as others that the Authors consider adequate to properly introduce previous results to the reader), the “Objective” of this paper can be now stated.  Quoting from the new submission, I suggest for instance something like:

“On the basis of the results obtained from our previous analysis, and in particular, Wang et al (1999, 2014), the objectives of this study are to:

(a) develop a NEW modified Green-Ampt model for water infiltration in fine textured soil with coarse interlayer, considering the unsaturated condition of the wetted zone; THE TESTING HYPOTHESIS IS THAT NEW MODEL MGA-2 IMPROVES THE LAYERED MODEL BY WANG ET AL (1999, called here after MGA-1) BY ACCOUNTING FOR THE DIFFERENT SATURATION VALUES BEHIND THE WETTING FRONT.

(b) use a Richards equation-based model and experimental results presented in WANG et al 2014 to investigate the effect of coarse interlayer’s buried depth on ware infiltration using THE NEW MGA-2 MODEL, that accounts for the saturation coefficients for calculating qe and Ke based on the simulation results;

(c) test the proposed new model MGA-2 with the infiltration experiments and Richards model  PRESENTED IN WANG ET AL 2014 (and not in this study!) to verify mga-2.

Note that what I wrote above is just an example of possible changes, and the authors are obviously free to adapt the text as they best prefer. The point is that the manuscript must be much clearer in this sense, and similar changes should be applied elsewhere in the manuscript, including the conclusions and the abstract.

Thank you very much for this advice.

In the revised manuscript, we had changed or added the relevant statement about the previous works (Experiment in the manuscript) and the current works (The new MGA-2 model in the manuscript) to make it clear what is novel and what is not in the introduction, materials and methods, conclusions, etc. See Line 70-82, 92, 139, 185, 337-338, 350-351, etc.

For the RE-Model, we (not including Wang) coded it using an implicit finite difference method with the MATLAB software (Line 170-175 in the manuscript). While Wang et al. (2014) used the HYDRUS-1D model to conduct the infiltration simulation.

Round  2

Reviewer 2 Report

The new manuscript has addressed some of the most critical aspects indicated in the previous round of review. I think that the references to previous works (Wang et al) has been correctly added. In my opinion, the manuscript could be accepted after a new round of moderate revisions is completed. Given that the authors are particularly fast (!) in revising the paper, I don’t think that this new revision would significantly delay its publication.

L 35: Much of the literature is based on fundamentally the same topic. This severely restricts the possibility that this new model could be used in other fields of research. I would suggest citing works from other fields that could broaden the applications of this new model, such as

·        Pedretti et al 2011 “Combining physical-based models and satellite images for the spatio-temporal assessment of soil infiltration capacity” Stochastic environmental research and risk assessment, 25-8 (1065-1075)

·        Pedretti et al 2012 “Slurry wall containment performance: monitoring and modeling of unsaturated and saturated flow” Environmental monitoring and assessment, 84-2 (607-624) (by the way, here the coarse-fine-coarse setup of the numerical model is very similar to the layered structure developed for the new analytical solutions)

·        Masetti et al (2016) “Impact of a storm-water infiltration basin on the recharge dynamics in a highly permeable aquifer”, Water resources management, 30-1 (149-165)

·        Blackmore et al 2018 “Evaluation of single-and dual-porosity models for reproducing the release of external and internal tracers from heterogeneous waste-rock piles” Journal of contaminant hydrology, 214 (65-74)

L156 Some authors think that the Richards equation (as written in Eq 14) may not apply in heterogeneous soils where preferential flow exists. Any thoughts here? Can the solution be extended to a multiporosity of multipermeability approach?

L167-172 It is great that the authors have developed their own MATLAB code for the solution of the Richards equation. I wonder why, given that HYDRUS-1D is free and resolves the same equation. More importantly, the authors did not indicate if the MATLAB code was verified. If it was done, I suggest that the authors add a figure comparing the solutions from this new code and a known one (e.g. Hydrus or SEEP/W) as supplementary information. If it was not verified, the authors should write it very clearly around these lines 167-172 and explain why they did not verify it.

Author Response

The new manuscript has addressed some of the most critical aspects indicated in the previous round of review. I think that the references to previous works (Wang et al) has been correctly added. In my opinion, the manuscript could be accepted after a new round of moderate revisions is completed. Given that the authors are particularly fast (!) in revising the paper, I don’t think that this new revision would significantly delay its publication.

1. L 35: Much of the literature is based on fundamentally the same topic. This severely restricts the possibility that this new model could be used in other fields of research. I would suggest citing works from other fields that could broaden the applications of this new model, such as

·        Pedretti et al 2011 “Combining physical-based models and satellite images for the spatio-temporal assessment of soil infiltration capacity” Stochastic environmental research and risk assessment, 25-8 (1065-1075)

·        Pedretti et al 2012 “Slurry wall containment performance: monitoring and modeling of unsaturated and saturated flow” Environmental monitoring and assessment, 84-2 (607-624) (by the way, here the coarse-fine-coarse setup of the numerical model is very similar to the layered structure developed for the new analytical solutions)

·        Masetti et al (2016) “Impact of a storm-water infiltration basin on the recharge dynamics in a highly permeable aquifer”, Water resources management, 30-1 (149-165)

·        Blackmore et al 2018 “Evaluation of single-and dual-porosity models for reproducing the release of external and internal tracers from heterogeneous waste-rock piles” Journal of contaminant hydrology, 214 (65-74)

We had added the relevant references to the revised manuscript. Line 32-35.

2. L156 Some authors think that the Richards equation (as written in Eq 14) may not apply in heterogeneous soils where preferential flow exists. Any thoughts here? Can the solution be extended to a multiporosity of multipermeability approach?

The Richards equation could not simulate the preferential flow in heterogeneous soils, where the dual-porosity model or the kinematic wave approach could be used to deal with it. Besides, for distinctly layered soil structure where fingering flow developed, the fingering phenomenon could not be depicted by the one dimensional Richards equation, as stated in our previous research (Wang et al., 2014), it is unable to describe the detailed fingering phenomenon and the associated solute transport. However, our previous study (Wang et al., 2014) also demonstrated that the Richards equation did show good performance of modeling layered soil columns (or to say, for one dimensional flow in multiporosity or multipermeability conditions). Also, Richards equation had been used by many scholars in the field scale under multiporosity or multipermeability conditions where the preferential flow was not eminent. We added more discussions in the manuscript. Line 158-161.

References:

Wang, C., Mao, X., Hatano, R., 2014. Modeling ponded infiltration in fine

textured soils with coarse interlayer. Soil Science Society of America Journal,

78(3): 745-753.

3. L167-172 It is great that the authors have developed their own MATLAB code for the solution of the Richards equation. I wonder why, given that HYDRUS-1D is free and resolves the same equation. More importantly, the authors did not indicate if the MATLAB code was verified. If it was done, I suggest that the authors add a figure comparing the solutions from this new code and a known one (e.g. Hydrus or SEEP/W) as supplementary information. If it was not verified, the authors should write it very clearly around these lines 167-172 and explain why they did not verify it.

In fact, this MATLAB code for solving the Richards equation is a part of our recently developed one-dimensional agro-eco-hydrological model. This model is capable of simulating water, solute and heat transport in layered soil coupled with crop growth, and the relevant manuscript is under review. The reason we use our own model instead of HYDRUS-1D is that it is more convenient to use our own model for pre- and post- processing.

A comparison was added for the solutions from the new code and the Hydrus-1D model for infiltration case (L1S1L1) to verify the new code in the revised manuscript. See Line 178-181, 206-209 and Figure 2.

This manuscript is a resubmission of an earlier submission. The following is a list of the peer review reports and author responses from that submission.

Round  1

Reviewer 1 Report

Dear authors,

I enjoyed reading your paper. I have a few comments, and I am looking forward to the publication!

Kind regards, your reviewer

Section 2: Please add a figure with the soil profile, showing the three layers 1, 2 and 3, and indicate the lengths L1, L2, L3 in the figure.
Line 86: According to the text, equation 1 is already the modified Green-Ampt-model. Do I understand this right? Please add the original equation from the Green-Ampt paper (1911) and the reference to make clear what the modifications are. I understand that the the modification with respect to the original model of Green & Ampt is the ability to handle three layers of soil. 
Line 88: Isn't it matrix, instead of matric? The word matric appears several times in the paper.
Line 119: Can you add a photograph of the infiltration experiment?
Line 152: "[...] can be obtained as simulation results" is wrong. The primary simulation results of the Richard equation are h and theta (the unknown variables in the equation) over time and space. Usually, secondary values (velocity, infiltration rates) are obtained from the solution as well. But the computation of cumulative infiltration and wetting front I would consider as post processing of simulation results. So consider to write: "can be obtained from simulation results by postprocessing".
Section 2.2: Please add some information about the way you solved the Richards equation. Has it been solved numerically? If so, did you code it by yourself or did you use an existing code like HYDRUS or FEFLOW? If so, add a reference to the software. Please add information about the grid resolution and the numerical scheme and the cell shape (e. g. one dimensional line elements or 2D triangles or rectangular cells).
Line 167: How did you identify the wetting front? I would expect something like the following: "nodes with a saturation of 0.99 or 1.0 are considered as saturated, others as unsaturated. The wetting front is the cell interface of two cells where one is saturated and the other one not."
Figure 1 and 2: The depth of the wetting front has not a straight course. There is nothing wrong with this, it is well known that solving the Richards equation is very challenging, but please discuss probable reasons. Is it due to the definition of the wetting front and the determination of its position (see above), or discretization, or numerical inaccuracy? The infiltration curves look more accurate. Figure 1 and 2: Add two horizontal lines that indicate the boundaries of the soil layers (assigned to the secondary axis). This helps the reader to understand the graphs.
Figure 2a: There is a jump in the infiltration curve at ca. t = 40 min. Does this coincide with the boundary of the top soil layer?
Line 193: It would make sense to me to add the bury depth of 22.5 cm to the list in order to connect chapter 3.2 to the previous chapter. It can also make sense to use the results of buried depth of 20 cm for Figures 1 and 2.
Figure 3: Add the results with buried depths of 22.5 cm to the graph.
Line 218: Typo: write "left" instead of "lest".
Line 240: Consider to write "is an approximation" instead of "is approach"
Line 245: Is the reference to Eq. 25-28 correct? I think it should be 25-27.
Figure 7: The course of the black line (LS3L) differs from the other two for depth greater than 50 cm. Please explain why. Is this due to initial conditions and boundary conditions?
Line 257: I do not follow why the abbreviation MAG-2 has been chosen. Wouldn't it make sense to use MGA instead? M for modified, G for Green and A for Ampt.
Line 259: Are the equation numbers correct? I think it should be 25 to 27.
Figure 10: Add the position of the interface between layer 1 and 2 as well as layer 2 and 3 to the graphs as a horizontal line.
Section 4: Write in present tense where appropriate. For line 303: indicate instead of indicated, and in line 305: becomes instead of became, etc.
Line 299: The model was not validated against the experiments themselves, but against the data from the experiments. Consider to write: "The [...] model was validated against the measured data from the ponded infiltration experiments ..."
Other: I think a discussion section should be added. The Green-Ampt model assumes a sharp wetting front (piston-shaped wetting front). Is this assumption feasible against the observations from the experiments?

Reviewer 2 Report

The authors propose a new version of the well known Green Ampt model to reproduce infiltration in unsaturated layered soils. While the manuscript fits the scope of Water, I don’t think that the manuscript should be published. The main reason is about the actual novelty of this manuscript. Most of the analysis and results are very similar (some of them identical) to what presented in Wang et al 2014 (doi: 10.2136/sssaj2013.12.053). This is only partially acknowledged in the manuscript, for instance around Lines 120 ,where the authors indicate that the setup is presented in that work. However, if one observes Figure 7 of Wang et al 2014 and Figure 1 of the present work, the fitted model results are basically identical. Other parts of the text (e.g. L202-203) remark that some findings were already identified in Wang et al 2014. The application of the modified Green-Ampt model, which should be the main aspect to emphasize in the paper, is presented and discussed only within a few lines in Section 3.3. This is clearly insufficient for the whole manuscript to be published. My recommendation is thus to reject it.